# Assessment of Cognitive Functions in Multimorbid Patients in Lithuanian Primary Care Settings: A Cross-Sectional Study Using MMSE and LT-GPCOG

**DOI:** 10.3390/medicina61010122

**Published:** 2025-01-14

**Authors:** Silvija Valdonė Alšauskė, Ingrida Grabauskytė, Ida Liseckienė, Jūratė Macijauskienė

**Affiliations:** 1Family Medicine Clinic, Lithuanian University of Health Sciences, LT-44307 Kaunas, Lithuania; 2Department of Physics, Mathematics and Biophysics, Lithuanian University of Health Sciences, LT-44307 Kaunas, Lithuania; 3Faculty of Public Health, Medical Academy, Lithuanian University of Health Sciences, Tilžės g. 18, LT-47181 Kaunas, Lithuania; 4Faculty of Nursing, Lithuanian University of Health Sciences, LT-44307 Kaunas, Lithuania

**Keywords:** cognitive impairment, multimorbidity, MMSE, GPCOG, COPD, CKD

## Abstract

*Background and Objectives*: The aging population has led to a rise in cognitive impairments, including dementia, often associated with multimorbidity. Early diagnosis of cognitive decline is crucial, especially in primary care, where time constraints and the limitations of diagnostic tools may hinder accurate detection. This study aims to assess the cognitive functions of multimorbid patients using the Mini-Mental State Examination (MMSE) and the Lithuanian version of the General Practitioner Assessment of Cognition (LT-GPCOG). We hypothesized that the LT-GPCOG would perform similarly to the MMSE in suspecting cognitive impairments. *Materials and Methods*: This cross-sectional study, conducted from 2021 to 2022, included 796 patients aged 40–85, with arterial hypertension and at least one other chronic disease, recruited from seven Lithuanian primary health care centers. Cognitive function was assessed using the MMSE and LT-GPCOG, and statistical analyses were performed using SPSS to determine the association between cognitive impairment and various demographic and clinical variables. *Results*: Out of 796 participants, 793 completed the study. Cognitive impairment was suspected in 5.1% of participants based on MMSE and 4.2% based on the LT-GPCOG. Statistically significant associations were found between cognitive impairment and chronic obstructive pulmonary disease (COPD) (*p* = 0.008 and *p* = 0.003) in both tests and chronic kidney disease (CKD) (*p* = 0.005) while testing with the MMSE. Lower education and unemployment were also correlated with cognitive impairment (*p* = 0.008 and *p* < 0.001). *Conclusions*: The findings suggest that regular cognitive assessments should be integrated into the management of multimorbid patients, particularly those with COPD and CKD. The LT-GPCOG proved to be an efficient alternative to the MMSE in primary care settings, demonstrating comparable diagnostic accuracy. Further studies are also needed to assess the sensitivity and specificity of the LT-GPCOG test.

## 1. Introduction

As life expectancy is growing worldwide, the aging of societies is a global phenomenon. It has long been known that the incidence of cognitive impairment increases with age, with the highest prevalence in persons aged 85 years and older (132 million people will suffer from dementia worldwide in 2050 [1]), making it a priority to develop instruments of early screening of at-risk individuals and interventions targeting prevention of loss of quality of life. Dementia affects approximately 55 million people worldwide (2021 estimate by WHO), and the number is expected to triple to 139 million by 2050 due to population aging. Dementia itself is a term for several diseases that are mostly progressive, affecting memory, other cognitive abilities, and behavior [1]. It impacts not only those affected, but also their families and the community in general, leading to increased health care costs and loss of productivity for economies [2]. Also, there is growing evidence that multimorbidity is also associated with cognitive impairment and dementia [3,4]

Misdiagnosis rates for dementia are high in primary care all over the world, and early detection of cognitive impairment could enhance treatment effectiveness and reduce costs [5]. The diagnosis of cognitive impairment and dementia in primary care is mostly based on clinical suspicion, which can be prone to error [6]. According to A. Bernstein et al.’s study conducted in in the USA in 2019, as many as two-thirds of people with dementia may be misdiagnosed in the primary care setting [7]. One of the main reasons is a lack of time. Primary care physicians often have limited time to spend with each patient, which can make it difficult to conduct a thorough cognitive assessment. This study highlights that the average primary care appointment lasts 15–20 min, often insufficient for a thorough cognitive assessment. Time constraints limit the use of comprehensive tools like the Mini-Mental State Examination (MMSE), increasing the risk of missed or delayed dementia diagnoses [7]. Also, primary care physicians may not have adequate training in diagnosing cognitive impairment, which can lead to misdiagnosis, especially when symptoms often overlap with other chronic conditions. Inadequate training can result in reliance on intuition or incomplete assessments, which in turn affect the quality of dementia care [8]. In addition, some medications can cause cognitive impairment, which can be mistakenly diagnosed as mild cognitive impairment [6]. Another important factor of misdiagnosing is that patients may be reluctant to report cognitive changes, which can delay diagnosis and treatment. Therefore, it is very important to involve a family member or caregiver in the diagnosis. Also, cognitive assessment for dementia ought to be equitable across cultures and educational backgrounds [9]. Certain tools, such as the MMSE, can be influenced by factors like the person’s age, education [10], socioeconomic status, and even symptoms of depression [11]. It is important to either modify total scores or employ distinct thresholds to accurately categorize individuals with limited education or those who are illiterate [12]. These considerations are among the key factors that prompted seeking an alternative tool capable of detecting cognitive impairment with greater sensitivity. In this context, the General Practitioner Assessment of Cognition (GPCOG) was selected as the cognitive screening instrument. The GPCOG is specifically designed to serve as an effective and practical tool for identifying cognitive impairment in primary care.

When it comes to a person’s socioeconomic status, education, or other factors that may affect test scores, it is important to note that the GPCOG, especially its informant part, has demonstrated lower levels of bias than the MMSE. The performance on the GPCOG seems to be independent of one’s cultural and linguistic background [13,14,15,16,17].

This study was designed to access cognitive functions of multimorbid patients using the MMSE and Lithuanian version of the General Practitioner Assessment of Cognition (LT-GPCOG) as a brief, efficient cognitive assessment instrument in Lithuanian patients with multimorbidity. The results of cognitive tests were analyzed in relation to socioeconomic indicators, as well as the number and nature of comorbidities. To address these constraints, there is a growing need for brief and efficient screening tools, such as the LT-GPCOG, which can deliver reliable results within a shorter time frame, better suited to the realities of primary care. Based on previous studies, we hypothesized that the LT-GPCOG will demonstrate equivalent sensitivity and specificity to the MMSE in detecting cognitive impairment in multimorbid patients, offering a viable alternative for time-constrained primary care settings [13,16,17].

## 2. Materials and Methods

### 2.1. Design and Participants

This cross-sectional study was part of a larger prospective experimental biomedical study project called “Improving health care for people with multiple chronic diseases and early detection of chronic diseases and their complications using the TELELISPA model”, project number 08.4.2-ESFA-K-616-01-0003. The main aim of the project was to improve the quality and availability of personal health care services for patients with two or more chronic non-infectious diseases. Data collection was conducted in 2021–2022. A total of 796 patients were included in this study, with 793 completing it. This biomedical study was performed at the Kaunas Clinic of the Lithuanian University of Health Sciences (LSMUL KK), the Family Medicine Clinic, and 6 other primary health care centers in Lithuania—3 urban and 3 rural. The participants were men and women, aged ≥40 to ≤85 years, diagnosed with arterial hypertension (ICD-10-AM codes I10, I11) and at least one other disease from the following:Diseases of the cardiovascular system (ICD-10-AM codes I50, I20, I25, I48).Diseases of the endocrine system (ICD-10-AM codes E11, E06, E89).Lung diseases (ICD-10-AM codes J44, J45).Diseases of the musculoskeletal system and connective tissue (ICD-10-AM codes M05, M06, M15-M19, M80, M81).

This study consisted of 793 participants. Demographic data for all subjects were collected, along with responses to questionnaires—GAD-7, PHQ-9, MMSE, GPCOG, and others. The cognitive functions of all patients were assessed at the end of this study. The MMSE and GPCOG questionnaires were administered on the same day, but not consecutively. This study was conducted by trained general practice nurses, advanced practice nurses, or family physicians.

### 2.2. Measures and Instruments

Demographic data such as age, gender, employment, and level of education were collected from all participants. Clinical and anamnestic findings (height, weight, BMI, waist circumference, BP, number of diseases, etc.) from the TELELISPA holistic questionnaire, laboratory test results, conclusions of instrumental tests, and data related to the course of treatment (used treatment, treatment complications) were collected from the TELELISPA subject group. Data of questionnaires filled out by patients were collected from all participants.

The MMSE was the first questionnaire used to assess cognitive function. It serves as a brief, objective screening tool for detecting cognitive impairment, estimating its severity, and tracking changes over time. The MMSE evaluates various cognitive domains, including registration/recall, calculation, attention, writing, and drawing. For example, patients may be asked to recall three words, perform serial subtractions, or draw intersecting pentagons. A score of 24 or below is considered abnormal, indicating potential cognitive impairment [18]. However, mild cognitive impairment patients may present with scores >24. The variation in MMSE cut-off scores across countries highlights the importance of accounting for cultural, educational, and socioeconomic contexts when interpreting cognitive screening results. Adjusting thresholds based on localized normative data ensures a more accurate assessment of cognitive impairment. The Lithuanian translation of the MMSE was prepared in collaboration with MiniMental LLC of Boston, MA, USA, which at the time had exclusive rights to the MMSE methodology. The translation was carried out using the double translation standard. The questionnaire has been used in practice unchanged since then.

The other cognitive assessment test was the General Practitioner Assessment of Cognition. GPCOG is a screening tool for cognitive impairment [14]. It is designed for general practitioners to detect cognitive impairment efficiently in primary care. It includes a patient cognitive test and an informant interview, enhancing predictive accuracy. The informant section is particularly useful for identifying subtle or early cognitive decline, providing insights into the patient’s daily functioning. The patient test comprises 9 questions, while the informant section has 6, and both can be scored separately or together [15]. The GPCOG can be completed in under 10 min, making it ideal for time-limited primary care settings and easy to integrate into routine visits. Permission for translation was obtained from the original author, and the Lithuanian version was developed through a multi-stage process: independent forward-translation by two translators, review by family physicians, and back-translation by two additional translators.

Similar to the initial version, the LT-GPCOG patient component consists of four cognitive exercises: assessing time orientation (date), evaluating visuospatial skills (clock-drawing test), testing episodic memory (recalling a recent news event), and assessing delayed recall (remembering a given name and address). Each correct answer scores 1 point, and higher scores reflect better cognitive performance (range 0–9). A score of 5–9 generally suggests that cognitive impairment is unlikely; nevertheless, a score ranging from 5 to 8 indicates that the patient needs informant evaluation. A score of 4 or below indicates possible cognitive impairment, warranting further evaluation. The informant segment comprises six inquiries regarding the patient’s present daily capabilities (such as recalling recent occurrences, remembering recent discussions, articulating thoughts fluently, handling financial matters and medications, and requiring aid with transportation) in contrast to their abilities from several years ago. Each question that is answered negatively scores 1 point, according to the sequential two-stage scoring method (range 0–6). A score of 0–3 indicates that cognitive impairment is unlikely. A score of 4–6 suggests potential cognitive impairment.

Translation of the statistical data was performed using ChatGPT-4o, an AI language model by OpenAI, based in San Francisco, CA, USA. The tool was used to translate Lithuanian content into English. The translations were reviewed and validated by the authors for accuracy.

### 2.3. Statistical Data Analyses

In this study, educational attainment was initially divided into four groups but was consolidated into three broader categories—(1) primary/secondary education; (2) specialized secondary, vocational, or college education; and (3) higher education—to increase statistical power and improve interpretability. Similarly, employment status was reduced from four categories to three: (1) employed (combining working and working pensioners), (2) unemployed, and (3) retired (including non-working pensioners). These adjustments ensured larger sample sizes, reducing error margins and enhancing the robustness of statistical analyses. Qualitative characteristics are described by presenting the frequency of their values and the relative frequency (in percentages). The statistical association of qualitative characteristics was examined using the method of crosstabs. Based on the table data, the chi-square (χ^2^) value and statistical significance (*p*-value) were calculated. For the description of qualitative variables, relative frequencies and percentages are provided. The chi-square test was used to determine the statistically significant difference in qualitative variables between groups—patients suspected of cognitive function impairment and patients without suspected cognitive function impairment.

To assess the likelihood of suspecting cognitive function impairment in patients with chronic diseases or other demographic indicators, univariate logistic regression analysis was performed. Some variables were not included in the statistical calculations due to the small number of participants.

IBM SPSS Statistics 29.0.1.0 (171) software was chosen for data processing. The influence of a variable was considered statistically significant when *p* < 0.05.

## 3. Results

A total of 793 patients completed this study. But only 754 patients’ data were suitable for further analysis. The demographic characteristics of the subjects who completed this study can be seen in Table 1.

The distribution of the number of chronic diseases among the subjects can be seen in Table 2.

Despite the arterial hypertension, which all the participants had, the most common chronic disease was angina pectoris—311 participants (41.2%) suffered from it. The smallest number of patients had post-procedural hypothyroidism—only 24 (3.2%). The incidence of other chronic diseases can be seen in Table 3.

Of all participants, only 751 completed the MMSE and only 692 completed the LT-GPCOG questionnaire. The distribution of participants can be seen in Table 4.

To examine a statistical association of qualitative characteristics, the crosstab method was used. Among all participants who underwent the MMSE, a statistically significant difference was observed only in few variables. Patients with COPD (J44) were statistically more likely to be suspected of cognitive impairment than those without suspected cognitive impairment (13.2% vs. 2.9%, respectively; *p* = 0.008). Similarly, patients with CKD (N18) were significantly more likely to be suspected of cognitive impairment compared to those without (18.4% vs. 5.2%, respectively; *p* = 0.005). Individuals not suspected of cognitive impairment based on MMSE scores were significantly less likely to have primary, secondary or vocational education or to have completed college than those who were suspected of cognitive impairment (22.2% vs. 36.8% and 48.2% vs. 55.3%; *p* = 0.008). Statistically, patients suspected of cognitive impairment were more likely to be unemployed or retired compared to those without suspected impairment (24.3% vs. 8.6% and 64.9% vs. 49.4%; *p* < 0.001). In terms of anxiety disorders assessed by GAD-7, individuals suspected of cognitive impairment had significantly higher instances of moderately severe and severe (16.2%) and moderate anxiety disorders (27.0%) compared to those not suspected of cognitive impairments (16.2% vs. 4.9% and 27.0% vs. 12.3%; *p* < 0.001). No statistically significant differences were observed among the remaining variables (such as cardiovascular, endocrinological, musculoskeletal diseases) between patients suspected of cognitive impairments and those not suspected (all *p* > 0.05)—all data can be seen in Table 5 and Table 6.

Among all participants tested with the GPCOG, a statistically significant difference was observed between suspected cognitive dysfunction and COPD (J44) and employment status. Patients with COPD were significantly more likely to be suspected of cognitive dysfunction than those not suspected (17.2% vs. 3.2%, respectively; *p* = 0.003). Among those suspected of cognitive dysfunction, 1 was employed (3.4%), 6 were unemployed (20.7%), and 22 were retirees (75.9%). Statistically, patients suspected of cognitive dysfunction were more likely to be unemployed or retired (20.7% vs. 8.8% and 75.9% vs. 49.1%, respectively; *p* < 0.001). No statistically significant differences were observed among the remaining variables (such as cardiovascular, endocrinological, or musculoskeletal diseases) in this group between patients suspected of cognitive dysfunctions and those not suspected (all *p* > 0.05)—all results can be seen in Table 7 and Table 8.

Upon conducting a univariate logistic regression analysis to predict the likelihood of suspecting cognitive function impairment, as assessed by the MMSE (MMSE ≤ 24) for all subjects, it was found that patients with COPD (J44) had a 4.993 times higher likelihood of being suspected of cognitive function impairment compared to those without COPD (*p* = 0.002). Patients suffering from CKD (N18), compared to those without this chronic condition, had a 4.126 times greater chance of cognitive function impairment suspicion (*p* = 0.002). Individuals who were unemployed at the time of this study, compared to those employed, had an 11.056 times higher likelihood of cognitive function impairment suspicion (*p* < 0.001). Moreover, patients who were already retirees had a 5.104 times greater likelihood of cognitive function impairment suspicion than those employed (*p* = 0.003). Due to a low number of cases, M80-81 were not included in the analysis. Education was also excluded due to statistical inaccuracies. The remaining variables were not statistically significant in predicting cognitive function impairment (all *p* > 0.05)—all data can be seen in Table 9.

Upon performing a univariate logistic regression analysis on all patients to predict the likelihood of suspecting cognitive function impairment assessed by the GPCOG questionnaire (GPCOGpatient score ≤ 4, or GPCOGinformant score ≤ 3), it was determined that patients with COPD had a 6.369 times higher chance of being suspected of cognitive function impairment compared to those without COPD (*p* < 0.001). Patients who were unemployed at the time of this study had a 28.500 times higher likelihood of cognitive function impairment suspicion compared to those who were employed (*p* = 0.002). Similarly, patients who were already retirees had an 18.817 times higher chance of cognitive function impairment suspicion than those employed (*p* = 0.004). Patients diagnosed with moderate anxiety disorder, according to the GAD-7 scale, compared to those with mild anxiety disorder, had a 3.150 higher likelihood of cognitive function impairment suspicion (*p* = 0.015). Due to a low number of cases, M80–81 were not included in the analysis. Education was also excluded due to statistical inaccuracies. The remaining variables in this group were not statistically significant in predicting cognitive dysfunction (all *p* > 0.05)—all data can be seen in Table 10.

## 4. Discussion

The findings from this study highlight several critical areas for further exploration and clinical consideration in the context of cognitive impairment among patients with chronic conditions. The results reveal associations between suspected cognitive dysfunction, chronic conditions such as COPD and CKD, and sociodemographic factors like lower education, unemployment, and retirement. These patterns were observed with both the MMSE and LT-GPCOG, highlighting the influence of chronic disease and socioeconomic status on cognitive health.

The significant association between COPD, CKD, and cognitive impairment suggests the need for integrated care approaches. The management of COPD and CKD should include regular cognitive assessments to identify early cognitive decline and provide timely interventions [19]. COPD patients warrant special attention because research indicates that cognitive impairment is a common comorbidity in these individuals, significantly affecting clinical outcomes and disease management [20]. Despite this, COPD remains an underrecognized risk factor for dementia [21].

Nevertheless, CKD patients often experience cognitive impairment due to shared anatomic and vasoregulatory features between the brain and kidneys, leading to cerebral hemodynamic changes. Other mechanisms, such as endothelial toxicity from the uremic state, also contribute. Additionally, early-stage research suggests involvement of purine nucleotides, oxidative stress, and Fibroblast Growth Factor 23 (FGF23). Understanding the complex interactions between kidney impairment and brain function and early detection is crucial for managing cognitive impairment in CKD patients [22].

The findings show a correlation between lower education levels and higher rates of suspected cognitive impairment. This may indicate that cognitive reserve, built through education, plays a protective role. Educational interventions and cognitive training could be beneficial in at-risk populations. One study suggested that more years of education was associated with higher cognitive level and slower cognitive decline in individuals with low or high educational attainment. The association between having more than 9 years of education and exhibiting slower cognitive decline was fully mediated by income [23].

Unemployment and retirement were linked to higher rates of cognitive impairment. This could reflect the impact of reduced mental engagement and social interaction on cognitive health. Encouraging continued mental and social activities in retired and unemployed individuals might help mitigate cognitive decline. There is strong evidence in the literature suggesting that unemployment negatively impacts cognitive function in both men and women. This aligns with several theoretical perspectives, including the “use-it-or-lose-it” hypothesis, which posits that mental activities regulate cognitive performance [24]. Additionally, the “cognitive reserve” hypothesis supports the idea that a mentally stimulating environment enhances neuronal resilience to pathological brain aging [25].

The MMSE, while widely utilized, may not serve as an optimal screening tool in this context, particularly when employing a cut-off value of 24. A higher cut-off, such as ≤26, may be more appropriate to capture cases of mild cognitive impairment. However, this threshold is likely influenced by educational attainment. Moreover, the cognitive impairments associated with conditions like COPD are typically characterized by deficits in frontal executive function rather than the amnestic subtype, for which the MMSE demonstrates limited sensitivity. Alternative assessments, such as the Montreal Cognitive Assessment (MoCA), which are more sensitive to frontal lobe dysfunction, may provide superior diagnostic accuracy, albeit at the cost of increased administration time.

Although the aim of this study was not to assess the cognitive function of patients with multiple morbidities, the results show the importance of timely diagnosis. None of the patients were asked about their cognitive complaints before they were recruited for this study, but patients with suspected cognitive impairment were still found. And although the numbers were not large, we can see that the GPCOG suspected cognitive impairment in a similar way as the MMSE. These results also do provide evidence supporting the use of the GPCOG in primary health care settings. The GPCOG saves time compared to the MMSE, making it more practical for time-limited primary care settings. Studies have shown that the GPCOG has similar diagnostic accuracy to the MMSE, confirming its effectiveness as a screening tool for dementia in primary care [16].

## 5. Conclusions

Suspected cognitive dysfunction was associated with COPD (13.2%, *p* = 0.008) while testing with the MMSE and COPD was significantly associated with suspected cognitive dysfunction (17.2%; *p* = 0.003) while testing with the LT-GPCOG. Suspected cognitive dysfunction was also associated with CKD (18.4%; *p* = 0.005) while testing with the MMSE. Lower education levels were more common among those suspected of cognitive dysfunction (36.8% of primary/secondary education and 55.3% of vocational, special secondary, or college education; *p* = 0.008) when tested with the MMSE. Unemployment and retirement were more frequent in the suspected group (24.3% were unemployed and 64.9% retired; *p* < 0.001) when testing with the MMSE, but similar results were while performing the LT-GPCOG—higher rates of unemployment and retirement were seen in those suspected of cognitive dysfunction (20.7% were unemployed and 75.9% retired; *p* < 0.001).

These findings emphasize the need to consider chronic diseases, education, and employment status in cognitive impairment assessments. Regular cognitive screening, particularly for patients with conditions like COPD or CKD, is essential for early detection and management. The LT-GPCOG’s shorter administration time makes it ideal for integration into routine primary care visits, allowing for cognitive screening to be streamlined alongside standard check-ups like blood pressure monitoring. This approach normalizes cognitive assessment, facilitates early detection of subtle declines, and minimizes disruptions to clinic workflows.

Further research is needed to explore the links between chronic diseases and cognitive impairment, including longitudinal studies to track cognitive decline and intervention effectiveness. Additionally, assessing the sensitivity and specificity of the LT-GPCOG compared to the MMSE is critical, as the MMSE may disadvantage individuals with lower education levels or diverse cultural and linguistic backgrounds. Methodological limitations include the lack of baseline cognitive complaint assessments and the absence of longitudinal follow-up. Future studies will address these gaps by categorizing participants based on cognitive complaints and monitoring changes over time.

## Figures and Tables

**Table 1 medicina-61-00122-t001:** Demographic characteristics of the subjects.

Main Characteristics		n	%
Age	45–50	46	6.0
	51–60	155	20.4
	61–70	266	35.0
	71–80	172	22.6
	80<	35	4.6
Gender	Men	294	39.0
	Women	460	61.0
Education	Primary or secondary education	172	23.0
	Specialized secondary, vocational education, or college education	362	48.5
	Higher education	213	28.5
Employment	Employed	289	40.4
	Unemployed	67	9.4
	Retired	360	50.2

**Table 2 medicina-61-00122-t002:** Number of chronic diseases among subjects.

Number of Chronic Diseases (Hypertension and Other Chronic Diseases)	n	%
2 chronic diseases	423	56.1
3 chronic diseases	171	22.7
4 chronic diseases	92	12.2
5 chronic diseases	51	6.8
6 chronic diseases	13	1.7
7 chronic diseases	4	0.5

**Table 3 medicina-61-00122-t003:** Most common chronic diseases without arterial hypertension.

Concomitant Chronic Diseases Without Arterial Hypertension (ICD-Code)	n	%
Angina pectoris (I20)	311	41.2
Chronic ischemic heart disease (I25)	77	10.2
Chronic heart failure (I50)	220	29.2
Atrial fibrillation and flutter (I48)	115	15.3
Type 2 diabetes (E11)	281	37.3
Autoimmune thyroiditis (E06.3)	88	11.7
Postprocedural hypothyroidism (E89)	24	3.2
Chronic obstructive pulmonary disease (J44)	26	3.4
Asthma (J45)	70	9.3
Rheumatoid arthritis (M05–M06)	48	6.4
Osteoporosis (M80–M81)	30	4.1
Chronic kidney failure (N18)	44	5.8

**Table 4 medicina-61-00122-t004:** Number and percentage of subjects suspected of cognitive impairment using the MMSE and LT-GPCOG.

	MMSE (n; %)	LT-GPCOG (n; %)
**Impairment of cognitive functions is not suspected**	713 (94.9)	663 (95.8)
**Suspected cognitive impairment**	38 (5.1)	29 (4.2)
**Total**	751	692

**Table 5 medicina-61-00122-t005:** Qualitative characteristics of diseases and MMSE scores: crosstab analysis.

Variable	Cognitive Impairment Not Suspected	Cognitive Impairment Suspected	*p* Value
I20, n (%):			1.000
Undiagnosed	420 (58.9)	22 (57.9)	
Diagnosed	293 (41.1)	16 (42.1)	
In total:	713	38	
I25, n (%):			0.263
Undiagnosed	643 (90.2)	32 (84.2)	
Diagnosed	70 (9.8)	6 (15.8)	
In total:	713	38	
I50, n (%):			0.210
Undiagnosed	509 (71.4)	23 (60.5)	
Diagnosed	204 (28.6)	15 (39.5)	
In total:	713	38	
I48, n (%):			0.753
Undiagnosed	605 (84.9)	31 (81.6)	
Diagnosed	108 (15.1)	7 (18.4)	
In total:	713	38	
E11, n (%):			0.251
Undiagnosed	451 (63.3)	20 (52.6)	
Diagnosed	262 (36.7)	18 (47.4)	
In total:	713	38	
E06.3, n (%):			0.194
Undiagnosed	632 (88.6)	31 (81.6)	
diagnosed	81 (11.4)	7 (18.4)	
In total:	713	38	
E89, n (%):			1.000
Undiagnosed	690 (96.8)	37 (97.4)	
Diagnosed	23 (3.2)	1 (2.6)	
In total:	713	38	
J44, n (%):			0.008
Undiagnosed	692 (97.1)	33 (86.8)	
Diagnosed	21 (2.9)	5 (13.2)	
In total:	713	38	
J45, n (%):			0.567
Undiagnosed	645 (90.5)	36 (94.7)	
Diagnosed	68 (9.5)	2 (5.3)	
In total:	713	38	
M05–M06, n (%):			0.504
Undiagnosed	666 (93.4)	37 (97.4)	
Diagnosed	47 (6.6)	1 (226)	
In total:	713	38	
M80–M81, n (%):			0.190
Undiagnosed with	686 (96.2)	35 (92.1)	
Diagnosed with	27 (3.8)	3 (7.9)	
In total:	713	38	
N18, n (%):			0.005
Undiagnosed	676 (94.8)	31 (81.6)	
Diagnosed	37 (5.2)	7 (18.4)	
In total:	713	38	

**Table 6 medicina-61-00122-t006:** Other qualitative characteristics and MMSE scores: crosstab analysis.

Variable	Cognitive Impairment Not Suspected	Cognitive Impairment Suspected	*p* Value
Gender, n (%):			0.651
Men	280 (39.3)	13 (34.2)	
Women	433 (60.7)	25 (65.8)	
In total:	713	38	
Education, n (%):			0.008
Primary, secondary	157 (22.2)	14 (36.8)	
Vocational/special secondary/college	341 (48.2)	21 (55.3)	
University	210 (29.7)	3 (7.9)	
In total:	708	38	
Employment, n (%):			<0.001
Employed	285 (42.0)	4 (10.8)	
Unemployed	58 (8.6)	9 (24.3)	
Pensioner	335 (49.4)	24 (64.9)	
In total:	678	37	

**Table 7 medicina-61-00122-t007:** Qualitative characteristics of diseases and LT-GPCOG scores: crosstab analysis.

Variable	Cognitive Impairment Not Suspected	Cognitive Impairment Suspected	*p* Value
I20, n (%):			0.293
Undiagnosed	380 (57.3)	20 (69.0)	
Diagnosed	283 (42.7)	9 (31.0)	
In total:	663	29	
I25, n (%):			0.218
Undiagnosed	595 (89.7)	24 (82.8)	
Diagnosed	68 (10.3)	5 (17.2)	
In total:	663	29	
I50, n (%):			0.706
Undiagnosed	468 (70.6)	19 (65.5)	
Diagnosed	195 (29.4)	10 (34.5)	
In total:	663	29	
I48, n (%):			1.000
Undiagnosed	559 (84.3)	25 (86.2)	
Diagnosed	104 (15.7)	4 (13.8)	
In total:	663	29	
E11, n (%):			1.000
Undiagnosed	415 (62.6)	18 (62.1)	
Diagnosed	248 (37.4)	11 (37.9)	
In total:	663	29	
E06.3, n (%):			0.774
Undiagnosed	581 (87.6)	25 (86.2)	
Diagnosed	82 (12.4)	4 (13.8)	
In total:	663	29	
E89, n (%):			0.616
Undiagnosed	642 (96.8)	28 (96.6)	
Diagnosed	21 (3.2)	1 (3.4)	
In total:	663	29	
J44, n (%):			0.003
Undiagnosed	642 (96.8)	24 (82.8)	
Diagnosed	21 (3.2)	5 (17.2)	
In total:	662	29	
J45, n (%):			0.741
Undiagnosed	603 (91.0)	26 (89.7)	
Diagnosed	60 (9.0)	3 (10.3)	
In total:	663	29	
M05–M06, n (%):			0.700
Undiagnosed	622 (93.8)	27 (93.1)	
Diagnosed	41 (6.2)	2 (6.9)	
In total:	663	29	
M80–M81, n (%):			1.000
Undiagnosed	637 (96.1)	28 (96.6)	
Diagnosed	26 (3.9)	1 (3.4)	
In total:	663	29	
N18, n (%):			0.687
Undiagnosed	624 (94.1)	27 (93.1)	
Diagnosed	39 (5.9)	2 (6.9)	
In total:	663	29	

**Table 8 medicina-61-00122-t008:** Other qualitative characteristics and LT-GPCOG scores: crosstabs analysis.

Variable	Cognitive Impairment Not Suspected	Cognitive Impairment Suspected	*p* Value
Gender, n (%):			0.763
Men	259 (39.1)	10 (34.5)	
Women	404 (60.9)	19 (65.5)	
In total:	663	29	
Education, n (%):			0.190
Primary, secondary	149 (22.6)	8 (27.6)	
Vocational/special secondary/college	316 (48.0)	17 (58.6)	
University	194 (29.4)	4 (13.8)	
In total:	659	29	
Employment, n (%):			<0.001
Employed	266 (42.0)	1 (3.4)	
Unemployed	56 (8.8)	6 (20.7)	
Pensioner	311 (49.1)	22 (75.9)	
In total:	633	29	

**Table 9 medicina-61-00122-t009:** Predicting cognitive impairment: univariate logistic regression based on MMSE scores.

	Odds Ratio	95% CI	*p* Value	AIC
I20 (vs. undiagnosed):				
Diagnosed	1.043	0.538–2.019	0.902	304.799
I25 (vs. undiagnosed):				
Diagnosed	1.722	0.696–4.262	0.240	303.574
I50 (vs. undiagnosed):				
Diagnosed	1.627	0.832–3.181	0.155	302.863
I48 (vs. undiagnosed):				
Diagnosed	1.265	0.543–2.946	0.586	304.530
E11 (vs. undiagnosed):				
Diagnosed	1.549	0.805–2.982	0.190	303.120
E06.3 (vs. undiagnosed):				
Diagnosed	1.762	0.751–4.131	0.193	303.288
E89 (vs. undiagnosed):				
Diagnosed	0.811	0.107–6.169	0.839	304.771
J44 (vs. undiagnosed):				
Diagnosed	4.993	1.772–14.068	0.002	297.849
J45 (vs. undiagnosed):				
Diagnosed	0.527	0.124–2.237	0.385	303.909
M05-M06 (vs. undiagnosed):				
Diagnosed	0.383	0.051–2.853	0.349	303.627
N18 (vs. undiagnosed):				
Diagnosed	4.126	1.704–9.990	0.002	297.055
Gender (vs. men)				
Women	1.244	0.626–2.471	0.534	304.420
Employment (vs. employed):				
Unemployed Pensioner	11.056 5.104	3.293–37.119 1.750–14.885	<0.001 0.003	277.264

**Table 10 medicina-61-00122-t010:** Predicting cognitive impairment: univariate logistic regression based on LT-GPCOG scores.

	Odds Ratio	95% CI	*p* Value	AIC
I20 (vs. undiagnosed):				
Diagnosed	0.604	0.271–1.347	0.218	243.163
I25 (vs. undiagnosed):				
Diagnosed	1.823	0.674–4.934	0.237	243.519
I50 (vs. undiagnosed):				
Diagnosed	1.263	0.577–2.766	0.559	244.427
I48 (vs. undiagnosed):				
Diagnosed	0.860	0.293–2.522	0.784	244.682
E11 (vs. undiagnosed):				
Diagnosed	1.023	0.475–2.201	0.954	244.757
E06.3 (vs. undiagnosed):				
Diagnosed	1.134	0.385–3.340	0.820	244.710
E89 (vs. undiagnosed):				
Diagnosed	1.092	0.142–8.409	0.933	244.753
J44 (vs. undiagnosed):				
Diagnosed	6.369	2.213–18.328	<0.001	236.097
J45 (vs. undiagnosed):				
Diagnosed	1.160	0.341–3.944	0.813	244.706
M05–M06 (vs. undiagnosed):				
Diagnosed	1.124	0.258–4.891	0.876	244.737
N18 (vs. undiagnosed):				
Diagnosed	1.185	0.272–5.167	0.821	244.711
Gender (vs. men)				
Women	1.218	0.558–2.661	0.621	244.511
Employment (vs. employed):				
Unemployed Pensioner	28.500 18.817	3.365–241.394 2.519–140.532	0.002 0.004	220.661

## Data Availability

Data is contained within the article.

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
