# Peer review of "Assessment of Cognitive Functions in Multimorbid Patients in Lithuanian Primary Care Settings: A Cross-Sectional Study Using MMSE and LT-GPCOG"

_medicina, 2025, doi:10.3390/medicina61010122_

Round 1

Reviewer 1 Report

Comments and Suggestions for Authors

The authors tested the cognitive status of patients presenting in primary care settings using the MMSE and GPCOG and found that they performed equally in identifying those with cognitive impairment. Chronic obstructive pulmonary disease and chronic kidney disease were the most important disorders associated with evidence of cognitive dysfunction. Since early detection of cognitive impairment may be important in the management of such patients. Such papers are welcomed. This paper is also important because it reminds us of the difficulty in identifying cognitive dysfunction in primary care settings, where time is not unlimited and does not allow the use of extensive and sensitive batteries of neuropsychological tests.

The paper is well written, statistics are adequate and references are appropriate.  

I have noticed some minor points.

Line 103. “Movement Disorders”. In neurology, the term is used for parkinsonian or cerebellar disorders. Although the ICD-10 numbers are provided, please clarify whether parkinsonian/cerebellar patients were included or not.

Line 123-124: “A score of 24 or below is considered abnormal, indicating potential cognitive impairment [18].” I would add a statement such as: However, in mild cognitive impairment patients may present with scores >23-24.

Line 169: in chi square, 2 should be superscript.

Discussion. MMSE although widely used may not be a good screening tool in this setting, at least with a cut-of value of 23 or 24. A cut of value 27 may be more appropriate in order to include cases with mild cognitive impairment, but this may be affected by educational status. Furthermore, the disorders described here (such as COPD) generally do not show the amnestic type of cognitive impairment, but rather the frontal one, for which MMSE shows low sensitivity. Tests more sensitive to frontal dysfunction such as the MoCA may be better (although slightly more time consuming). Please add such comments in the discussion.

Author Response

1) Line 103. “Movement Disorders”. In neurology, the term is used for parkinsonian or cerebellar disorders. Although the ICD-10 numbers are provided, please clarify whether parkinsonian/cerebellar patients were included or not.

Thank you for your note, it is a translation error. We have corrected it to "Diseases of the musculoskeletal system and connective tissue" as it refers to arthropathies, dorsopathies, osteopathies and other pathologies of the musculoskeletal system

2) Line 123-124: “A score of 24 or below is considered abnormal, indicating potential cognitive impairment [18].” I would add a statement such as: However, in mild cognitive impairment patients may present with scores >23-24.

Thank you for this comment. This is a very interesting place for a discussion. In Lithuania, mild cognitive impairment is suspected when the MMSE is in the range of 21-24 points. The authors, who have chosen this gradation, argue that the lower scores in post-Soviet countries were chosen because of educational gap and other peculiarities of that period (for examples Italians are using 26 as a cut-off point). This is why we started looking for a new instrument, because the MMSE no longer fulfils all the needs, not only at the primary level, but also for the next generation who did not suffer from the same things the older generation did. However, I will add a clarifying sentence, thank you very much for reading and noticing so carefully!

3) Line 169: in chi square, 2 should be superscript.

Thank you for noticing that!

4) Discussion. MMSE although widely used may not be a good screening tool in this setting, at least with a cut-of value of 23 or 24. A cut of value ≤ 27 may be more appropriate in order to include cases with mild cognitive impairment, but this may be affected by educational status. Furthermore, the disorders described here (such as COPD) generally do not show the amnestic type of cognitive impairment, but rather the frontal one, for which MMSE shows low sensitivity. Tests more sensitive to frontal dysfunction such as the MoCA may be better (although slightly more time consuming). Please add such comments in the discussion.

Thank you, I have taken your comment into consideration, adapted it slightly and added it to the discussion section. Thank you for putting so much time and work into reading this article

Reviewer 2 Report

Comments and Suggestions for Authors

This is a well-written paper assessing the cognitive functions of multimorbid patients using the Mini-14 Mental State Examination (MMSE) and the Lithuanian version of the General Practitioner Assessment of Cognition (LT-GPCOG).

1) Will everyone know what GPCOG stands for? The introduction did not clarify it the first time.

2) please mention the prevalence of cognitive impairments in the introduction 

3) there are some minor grammatical errors in the text. check again. 

4) The discussion should start with a short paragraph about main findings of the study. 

5) please also use references from 2024. 

Comments on the Quality of English Language

There are some minor grammatical errors 

Author Response

1) Will everyone know what GPCOG stands for? The introduction did not clarify it the first time.

Thank you for noticing. I have added an explanatory sentence

2) please mention the prevalence of cognitive impairments in the introduction 

Thank you for noticing that – I’ve added the prevalence of dementia in the introduction.

3) there are some minor grammatical errors in the text. check again.

Thank you for noticing that, I will check everything again 

4) The discussion should start with a short paragraph about main findings of the study. 

Thank you, I've added a sentence summarising the results at the beginning of the discussion.

5) please also use references from 2024. 

I sincerely regret that I was unable to address this comment. The article was initiated in 2024, at which time no adequate sources from that year were available; therefore, earlier references were utilized. The transition to a new calendar year in recent days may render these sources a bit outdated. However, at the time of writing, the article was based on the most recent and relevant sources available.